# Patient preferences for incentives in Contingency Management interventions in methadone treatment: A best-worst scale analysis

Thuy Thi Dieu Dao[1,2]*, Hue Thi Nguyen[1], Trang Thu Nguyen[1], Thuyet Thi Phung[3], Van Hai Hoang[1,4], Huong Thi Le[4], Brian W. Pence[2], Giang Minh Le[1,4], Vivian F. Go[5], William C. Miller[2]

1 Center for Training and Research on Substance Abuse and HIV, Hanoi Medical University, Hanoi, Vietnam, 2 Department of Epidemiology, University of North Carolina-Chapel Hill, Chapel Hill, North Carolina, United States of America, 3 Hanoi Center for Disease Control and Prevention, Hanoi, Vietnam, 4 Institute of Preventive Medicine and Public Health, Hanoi Medical University, Hanoi, Vietnam, 5 Department of Health Behavior, University of North Carolina-Chapel Hill, Chapel Hill, North Carolina, United States of America

* daodieuthuy@hmu.edu.vn

## Abstract

### Background

Contingency management (CM) effectively enhances adherence and retention in methadone maintenance treatment (MMT). But implementing CM in resource-limited settings is challenging, particularly due to costs associated with providing incentives. In this study, we aimed to describe and quantify patient preferences regarding low-cost CM incentives to promote adherence and retention in MMT.

### Methods

We conducted a cross-sectional survey using a best-worst scale (case 1) among 216 participants ages 18 or older undergoing MMT in six clinics in Hanoi, Vietnam. The study asked participants to complete 13 sets of best-worst scaling tasks. Each task presented a subset of four incentives chosen from a total of 13 incentives. Net scores for each incentive were calculated by subtracting the total times an incentive was rated as least appealing from the total times it was rated as most appealing. Standardized scores were derived by dividing the net score by the sum of selections and then converted to weighted probabilities (WP) that ranged from 0% to 100% (example interpretation: an incentive with WP of 20% is twice as desired as an incentive with WP of 10%). The 95% confidence intervals (95% CI) were estimated using bootstrapping.

**Data availability statement:** All relevant data are compressed in the Supporting information folder (S2 Data).

**Funding:** Research reported in this publication was supported by the Fogarty International Center, Eunice Kennedy Shriver National Institute of Child Health & Human Development, the National Institute on Drug Abuse, and the National Institute of Mental Health of the National Institutes of Health under Award Number D43 TW011548. The content is solely the responsibility of the authors and does not necessarily represent the official views of the National Institutes of Health The funders had no role in study design, data collection and analysis, decision to publish, or preparation of the manuscript.

**Competing interests:** The authors have declared that no competing interests exist.

## Results

The mean age of participants was 44.7 (SD = 8.0, range: 25–66). Most were male (95%), married (59%), and had not completed high school (69%). About half (50%) had been on methadone treatment for more than five years. The most preferred incentives were "discount for monthly methadone fees" (WP = 16.9, 95% CI: 16.0, 17.8) and "take-home methadone privileges" (WP = 11.3, 95% CI: 10.1, 12.6), followed by "priority coupons for early medical examinations/consultations". In contrast, the least preferred incentives were "being recognized/praised in their community" (WP = 4.5, 95% CI: 4.0, 5.0) and "being recognized/praised at their clinic" (WP = 4.7, 95% CI: 4.1, 5.4).

## Conclusions

Treatment fee support, take-home methadone privilege, and coupons for prioritizing checkup at clinics emerged as the most desirable incentives for patients. We recommend future CM intervention may consider using these incentives as the first-line rewards to offer to reinforce treatment adherence and retention in methadone treatment. These findings suggest potential low-cost CM strategies that could inform decision-making in MMT programs.

## Introduction

Substance use, mainly heroin use, remains an important driver of the HIV epidemic in Vietnam [1,2]. To address this issue, Vietnam launched the methadone maintenance treatment (MMT) program [3]. Up to the end of 2023, the MMT program has served nearly 51,000 people who use drugs (PWUD) at 343 methadone clinics in 63 provinces of the country [4]. In 2008, when MMT was first piloted in Vietnam, all methadone-related fees including consumables, services, and medication were fully covered by sponsors, making treatment free to patients. From 2015 onwards, financial responsibility for MMT shifted partially to provinces, leading them to collect medication fees from PWUD [5]. People using MMT began paying out-of-pocket at about 300,000 VND (equivalent to 12 US dollars) per month; at the time, the amount accounted for 5–10% of patients' income on average [5]. As a large proportion of individuals enrolled in MMT reported being unemployed or having unstable income sources [6,7], these economic vulnerabilities presented challenges for the individuals to sustaining retention in the program.

Of noted, MMT effectively reduces opioid use [8,9], and improves overall health and well-being among people who use drugs (PWUD) [10,11]. In addition, MMT decreases risk behaviors for HIV infection and crime [10]. Despite many efforts in Vietnam to increase MMT coverage, fewer than 50% of PWUD are undergoing treatment [12]. Furthermore, retention and adherence are low among PWUD on MMT [13,14]. Factors associated with low adherence and retention in MMT have been identified at both individual and structural levels. In addition to individual factors such

as older age, lower education, and comorbid health conditions, key structural barriers include stigma, and scheduling conflicts between clinic service hours and patients' availability [13]. These structural challenges are among the most significant factors to sustain adherence and retention. Given adherence and retention are key to MMT effectiveness, interventions are needed to ensure PWUD are adequately treated.

One evidence-based approach to increasing MMT adherence and retention is contingency management (CM). CM is based on the theory of operant conditioning, in which incentives are used to reinforce positive target behaviors [15]. CM has increased drug abstinence and treatment attendance in MMT settings [16,17]. But many questions about the benefit of CM for MMT retention and adherence remain [16]. First, an important issue is that previous studies has focused on drug abstinence as the target or rewarded behaviors which were essentially mirroring methadone treatment effects [18,19]. It is unclear whether other behaviors such as treatment adherence and retention, which are strongly linked to not only abstinence but long-term health outcomes, might be more appropriate targets for CM [20]. Second, the benefit of CM has been seen in high and low/ middle income countries [18,21,22]. But the use of CM in low/ middle income countries raises particular challenges which may hinder its effectiveness [21]. Therefore, we need to explore ways to strengthen the implementation of CM to ensure its effectiveness in low-income, real-world settings.

A major challenge of implementing CM in low/ middle income countries is the high cost associated with providing incentives [21]. In addition, low willingness to implement CM and substance use-related stigma present challenges at the contextual and cultural levels [21]. These barriers limit the implementation of CM for MMT in resource limited settings. To address these barriers to strengthen the implementation, some upstream implementation outcomes, such as acceptability and feasibility, were particularly appropriate to focus on at the beginning, which may include cost-reduced and more acceptable/ appealing incentives. Furthermore, based on findings from our scoping review (publication currently under review), CM is highly adaptable across different settings. Again, a key factor in successful CM implementation is identifying the types of incentives that are most motivating for the target population. In other words, rewards must be meaningful to patients in order to be effective. In the MMT clinic settings, a critical gap remains in understanding the preferences of PWUD about specific types of CM incentives, particularly given the need for cost-effective solutions [23]. Taken together, there is a clear need to explore the incentive options which are both motivating for patients and feasible for implementation at the policy level.

We conducted the study to address this gap in understanding PWUD preferences. Here, we describe and quantify patient preferences regarding low-cost incentives in CM packages to promote adherence and retention in methadone treatment.

## Methods

### Study design

We conducted a survey among 216 participants receiving methadone treatment in three clinics in the city center and three clinics in the suburban areas in Hanoi, Vietnam (including MMT Dong Anh, Dong Da, Hai Ba Trung, Nam Tu Liem, Ung Hoa and Dan Phuong clinics). Data collection was conducted from November 14 to November 26, 2023, with one day of data collection at each study site. The study used a case 1 best-worst scaling (BWS) approach ("object" case) to evaluate people's preferences for CM incentives [24]. BWS is a ranking approach in which respondents repeatedly choose the two objects in varying sets of three or more objects that they find best and worst on an underlying continuum of interest [24]. In our study, participants chose the most and the least appealing incentives out of a set of four incentives, repeated 13 times with varying incentives included each time. We used 13 types of incentives in total; each incentive appeared 4 times in the entire set for each person.

### Setting

This study was conducted in Hanoi, one of the provinces with the largest number of people who use drugs in Vietnam. In 2025, the province provided methadone treatment for about 3,800 PWUD at 22 methadone clinics [25].

## Participants

The eligibility criteria included: (1) ≥18 years of age, (2) currently enrolled in one of the six methadone clinics, (3) provision of written informed consent. At the beginning of data collection phase, the study team directed clinical staff to prioritize approaching patients who presented poor adherence to methadone treatment or had a history of discontinuation of methadone. On data collection days, clinical staff, based on their knowledge of their patients, approached patients as they showed up at the clinic, then referred interested patients to research assistants, who then screened the eligible criteria, introduced the study information, obtained informed consent, and administered the survey. Although poor adherence or previous discontinuation of methadone was not an explicit inclusion criterion, the target population we aimed to reach out consisted of patients currently receiving methadone who demonstrated poor adherence and/or were at high risk of treatment discontinuation. These aspects were target behaviors for the CM interventions. No exclusion criteria were applied other than inability to understand the study procedure and refusal to participate.

The study proposal was reviewed and approved by the Institutional Review Board (IRB) at Hanoi Medical University (number: IRB-VN01.001/ IRB00003121/ FWA00004148). All participants provided written consent and received 50,000 Vietnam Dong (equivalent to 2 U.S dollars) as compensation for transportation upon completing the survey. All information of the participants was collected confidentially by research assistants of the study team (i.e., the participant's names and medical record identifiers were not collected).

## Questionnaire development and data collection

To identify potential types of low-cost incentives for a CM package aimed at improving adherence and retention in MMT, we conducted a scoping review that included 14 scientific articles (publication in press). From this review, we identified 12 commonly used incentive types reported in the CM literature (Table 1, excluding the 12th incentive). In which, six types of incentives are non-monetary including take-home methadone privilege, receiving encouraging text messages daily/ weekly, priority coupons for early checkup or counseling, recognition or praise at the methadone clinic, job recommendation, and referral to health care services. The remaining six types of incentives are considered low-cost and include options such as vouchers that can be exchanged for goods at grocery stores or supermarkets.

Following the scoping review, we carried out formative qualitative research, which involved 12 focus group discussions (FGDs) separately with a total of 40 patients and 40 MMT clinical providers from six methadone clinics (three located in

**Table 1. List of thirteen types of incentives.**

| Incentive code | Incentive description |
|---|---|
| 1 | Cash |
| 2 | Vouchers to exchange goods in grocery stores or supermarkets |
| 3 | Vouchers for discounting in health care services (e.g., fees for clinical tests or checkup) |
| 4 | Lottery tickets for a chance to win prizes |
| 5 | Take-home methadone privilege |
| 6 | Receiving encouraging text messages daily/ weekly |
| 7 | Support or discount for methadone monthly fees |
| 8 | Priority coupons for early checkup or counseling |
| 9 | Recognition or praise at the methadone clinic |
| 10 | Job recommendation and introduction |
| 11 | Referral to health care services |
| 12 | Recognition or praise at the patient's community |
| 13 | In-kind gifts for the patient's family |

urban areas and three in suburban areas of Hanoi). These clinics also served as the study sites for the current paper. The qualitative findings confirmed the 12 incentive types identified in the scoping review and additionally suggested one more incentive: recognition or praise within the patient's community (Table 1).

We then developed a structured questionnaire with appropriate layout for best-worst scale tasks based on 13 specified incentives [24]. Ultimately, the questionnaire for this paper consists of two parts: participants' socio-demographic characteristics and best-worst scale exercises with 13 sets of best-worst scaling tasks.

Data collection was carried out through face-to-face interviews using the Kobo Toolbox platform. First, patients in methadone treatment at the study sites as potential participants were approached by clinical staff and asked verbally if they wanted to consider participating in the study. Then if the patients agreed, the clinical staff referred to two research assistants to screen eligibility and introduce the study information to the patients. Upon the patients met the eligibility criteria and agreed to participate in the study, they signed a written consent form provided by research assistants. Two research assistants were trained by the research team.

### Measures

We measured patient preferences regarding incentives in CM packages using Case 1 BWS as described above. Participants were asked to complete 13 sets of best-worst scaling tasks. Each task presented a subset of four types of incentives chosen from a total of 13 incentives (Table 1). For each task, participants identified the most and the least appealing incentives. Since each question will derive two selections/objects (most and least appealing), 13 sets of questions will result in a total of selections equal to the sample size multiplied by 13 sets and 2 objects (i.e., 216 * 13 * 2 = 5616 total selections). We also asked participants about methadone treatment information, such as date of treatment initiation, history of dropout, missing doses, and sociodemographic characteristics, including age, sex at birth, education, employment status, and marital status.

### Statistical analysis

We described our study sample characteristics using frequency and proportion for qualitative variables and mean and standard deviation (STDEV) for continuous variables. We did not fit any univariable or multivariable regression analyses, as our focus was purely descriptive and not aimed at assessing associations and controlling confounders. One potential source of bias was reporting bias, which we sought to minimize by thorough training and involving experienced interviewers. We reported missing data using counts, which was not likely a big issue for our study focus.

For the best-worst scale analysis, we estimated and reported net scores, standardized scores (i.e., relative preference score) and weighted probabilities (i.e., weighted preference score) for each of the 13 incentives. Net scores for each incentive were calculated by subtracting the total times an incentive was rated as least appealing from the total times it was rated as most appealing. Then using the calculated net scores for each incentive, we derived standardized scores by dividing the net score by the sum of selections and then converted to weighted probabilities (WP) that ranged from 0% to 100% (e.g., an incentive with WP of 20% is twice as desired as an incentive with WP of 10%). The 95% confidence intervals (95% CI) were estimated using the bootstrap method with resampling 1000 times. Data analyses were performed using RStudio (version: 2024.09.0−375).

### Results

We enrolled a total of 216 participants. Following verbal approach and referral by clinical staff, 232 patients met with research assistants. After research assistants provided a full introduction of the study, approximately 7% of patients refused to participate in the study (S1 Table), which derived a final sample of 216 participants. The main reasons for non-participation included work obligations, lack of time (e.g., unable to wait for the survey interview), and household

responsibilities (e.g., need to return home to care for a grandchild). We did not collect data on the total number of patients verbally approached by clinical staff at the study sites.

The sample had relatively similar characteristics across the six MMT clinics (Table 2). The mean age of participants was 45 years (STDEV = 8.0, range: 25–66). Most study participants were older than 40 (71%), had unstable paid jobs

**Table 2. Socio-demographic and methadone treatment characteristics of participants (N = 216).**

| Characteristics | Frequency (%) |
|---|---|
| Study sites | |
| MMT clinic 1 (urban) | 50 (23) |
| MMT clinic 2 (urban) | 53 (24) |
| MMT clinic 3 (urban) | 36 (17) |
| MMT clinic 4 (suburb) | 27 (13) |
| MMT clinic 5 (suburb) | 20 (9) |
| MMT clinic 6 (suburb) | 30 (14) |
| Age group* Mean (STDEV): 45 (8.0) years | |
| ≤ 30 | 6 (3) |
| > 30 - ≤ 40 | 56 (26) |
| > 40 - ≤ 50 | 98 (46) |
| > 50 | 55 (26) |
| Employment status | |
| Stable jobs | 48 (22) |
| Unstable jobs | 141 (65) |
| Unemployed | 27 (13) |
| Education level | |
| Less than high school | 150 (69) |
| High school or above | 66 (31) |
| Sex at birth | |
| Male | 205 (95) |
| Female | 11 (5) |
| Marital status | |
| Married/living with a partner | 127 (59) |
| Widowed/divorced/separated | 45 (21) |
| Never married | 44 (20) |
| MMT duration by group Mean (STDEV): 5 (3.6) years | |
| ≤ 1 year | 41 (19) |
| > 1 year to ≤ 5 years | 68 (32) |
| > 5 years | 107 (49) |
| Ever dropped out from MMT | |
| Never | 169 (78) |
| Yes | 47 (22) |
| Missed MMT dose during the past 30 days | |
| No, no missing doses | 155 (72) |
| Yes | 61 (28) |

STDEV: standard deviation; MMT: methadone maintenance treatment; *1 missing value; **43 missing months of MMT initiation (assumed 99 = June)

(65%), and were married or living with a partner (59%). Most were male (95%) and had not completed high school (69%). About half (50%) had been on methadone treatment for more than five years, and around 22% had a history of dropping out from methadone treatment. Around one-third reported missing methadone doses during the past 30 days before interview.

The most preferred incentive was "discount for monthly methadone fees" (rank 1st with WP = 16.9, 95% CI: 16.0, 17.8; Figs 1 and 2, Table 3). The second most preferred incentive was "take-home methadone privileges" (WP = 11.3, 95% CI: 10.1, 12.6), followed by "priority coupons for early medical check-up or consultations" (WP = 9.6, 95% CI: 8.5, 10.8). In contrast, the least preferred incentives were "being recognized and praised in their community" (WP = 4.5, 95% CI: 4.0, 5.0) and "being recognized and praised at their clinic" (WP = 4.7, 95% CI: 4.1, 5.4).

## Discussion

Our findings highlight the preferences of people with OUD for incentives in contingency management interventions to enhance adherence and retention in outpatient methadone clinical settings. We found that financial support or discounts for monthly treatment fees were the most desirable incentives followed by take-home methadone privileges. The least preferrable incentives were recognition or praise in the patient's community or in the methadone clinics.

In Vietnam, since the launch of the MMT program in 2008, which was initially free of charge until the introduction of co-payments in 2015, many studies have reported that the methadone fees pose challenges to patient retention in the program. Inability to afford methadone monthly fees [6], experience or awareness of hardship in co-paying [5], and economic vulnerability have been associated with increased dropout from methadone among Vietnamese people using MMT

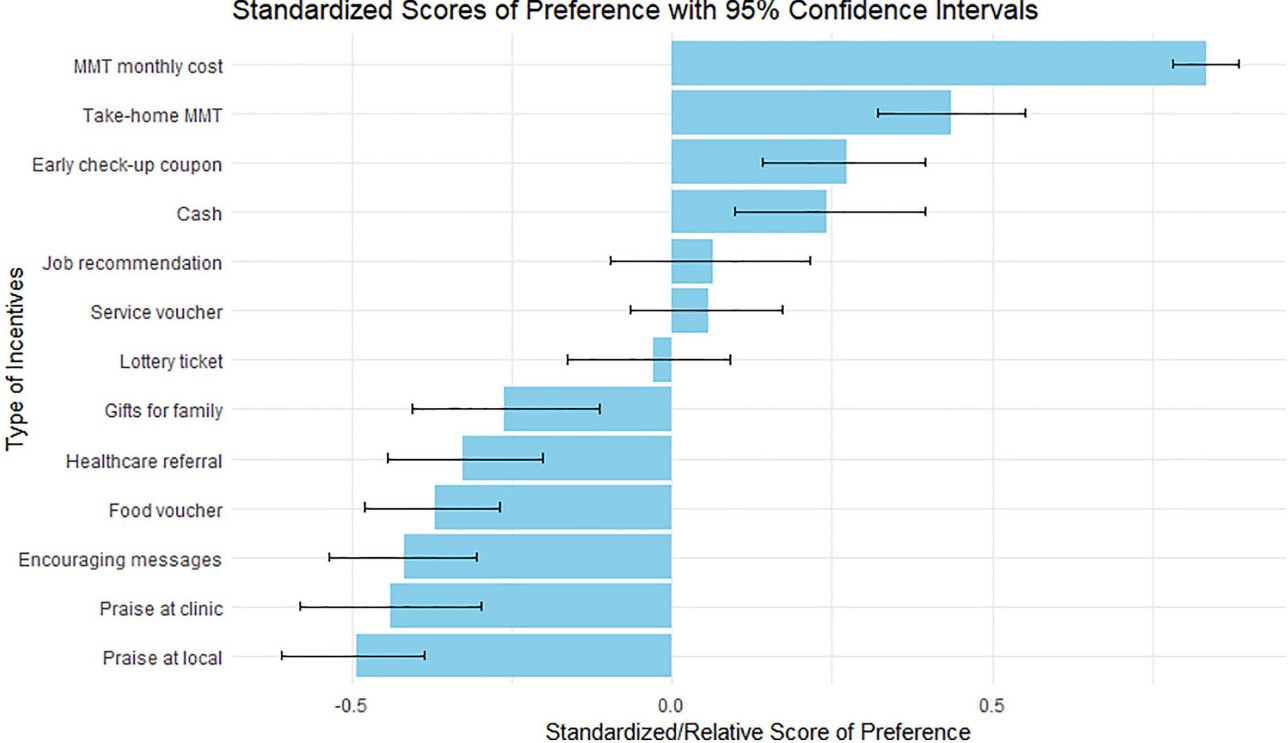

**Fig 1. Standardized score of preference in each contingency management incentive.** Blue bars show the value of standardized scores. Black lines across the bars present 95% confidence intervals of the standardized score using bootstrap resampling 1000 times.

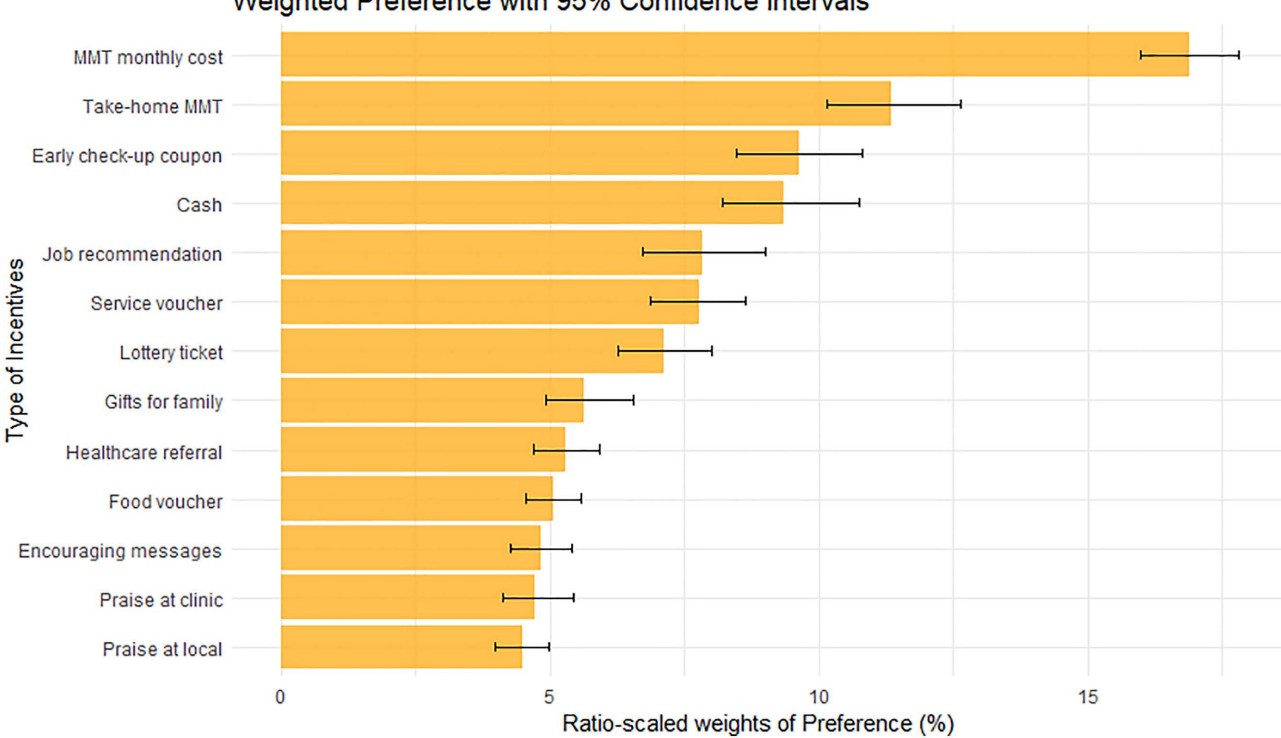

**Fig 2. Ratio-scaled weights of preferences.** Orange bars show the value of ratio-scale weighted scores (ranging from 0% − 100% for weighted scores). Black lines across the bars present 95% confidence intervals of the weighted score using bootstrap resampling 1000 times.

**Table 3. Best-worst scaling estimates and ranking preferences in contingency management incentives.**

| Incentive code | Description | Best count | Worst count | Net score | Standardized/Relative score (95% CI) | Weighted preference score (95% CI) | Rank |
|---|---|---|---|---|---|---|---|
| 7 | Support for the monthly cost of MMT | 534 | 48 | 486 | 0.84 (0.78, 0.88) | 16.90 (15.98, 17.80) | 1 |
| 5 | Take home MMT privileges | 423 | 166 | 257 | 0.44 (0.32, 0.55) | 11.34 (10.14, 12.62) | 2 |
| 8 | Get priority coupon for early medical check-ups or early consultation | 219 | 125 | 94 | 0.27 (0.14, 0.39) | 9.64 (8.46, 10.80) | 3 |
| 1 | Cash | 243 | 148 | 95 | 0.24 (0.10, 0.39) | 9.35 (8.20, 10.75) | 4 |
| 10 | Job recommendations | 220 | 193 | 27 | 0.07 (−0.10, 0.22) | 7.83 (6.72, 9.01) | 5 |
| 3 | Accumulated vouchers for services (testing or examination fees) | 220 | 196 | 24 | 0.06 (−0.07, 0.17) | 7.77 (6.88, 8.65) | 6 |
| 4 | Receive a lottery ticket for a chance to win prizes | 190 | 201 | −11 | −0.03 (−0.16, 0.09) | 7.13 (6.28, 8.01) | 7 |
| 13 | In-kind gifts for their families (parents, children) | 143 | 244 | −101 | −0.26 (−0.40, −0.11) | 5.65 (4.91, 6.55) | 8 |
| 11 | Healthcare service linkage and referral | 118 | 232 | −114 | −0.33 (−0.44, −0.20) | 5.29 (4.70, 5.93) | 9 |
| 2 | Accumulated vouchers exchange for goods | 150 | 326 | −176 | −0.37 (−0.48, −0.27) | 5.07 (4.54, 5.57) | 10 |
| 6 | Receive encouragement messages daily/weekly | 102 | 248 | −146 | −0.42 (−0.54, −0.30) | 4.83 (4.26, 5.42) | 11 |
| 9 | Recognized and praised at the clinic | 107 | 274 | −167 | −0.44 (−0.58, −0.30) | 4.73 (4.13, 5.44) | 12 |
| 12 | Recognized/praised at their community | 139 | 407 | −268 | −0.49 (−0.61, −0.39) | 4.49 (3.97, 4.97) | 13 |

MMT: methadone maintenance treatment; 95% CI: 95% confidence interval

[7]. Our findings are consistent with the literature, our participants' employment situations involved unstable paid jobs or unemployment. Certain subgroups of patients, such as those with comorbidities and low socioeconomic status, are not only at higher risk of poor adherence to MMT [13] but also show lower willingness to pay for methadone treatment fees [26], making them more vulnerable to treatment discontinuation. Thus, considering a support for monthly treatment fees, particularly for some disadvantaged subgroups, could serve as both an appealing and practical incentive to maintaining people on MMT.

We also found that take-home methadone and priority coupons for early checkups or counselling were also among the top preferences. These recovery-oriented incentives were more desirable than other monetary incentives (e.g., cash, coupon for goods), presumably because the recovery-oriented incentives would help transition to stability. Stability was desirable, as half of our study participants had been on methadone more than 5 years. Longer duration in methadone has been associated with lower health-related quality of life [27], increased treatment fatigue [28], and increased rates of dropout [29,30]. Furthermore, a common reason for quitting methadone has been conflict with work [6]. Prioritization for early checkups reduces waiting times, which allows people using MMT to minimize job disruption. Take-home methadone appears to be effective [31], and in addition, enhances chances to secure jobs and improving financial stability. In Vietnam, regulatory and operational challenges in the take-home methadone program have resulted in restricting this option to people with a history of good adherence. Given the low cost of these recovery-oriented incentives, this is particularly relevant for the cost-effective implementation of CM in Vietnam and other low- and middle-income countries.

Receiving recognition or praise in the patient's community was the least preferable incentive. PWUD in Vietnam experience high levels of stigma and discrimination which often persist even if drug use has been reduced or treatment has started [28,32]. This stigma may lead to poor health outcomes and treatment dropout [33]. Recognition or praise-based CM shares principles with 12-step substance use treatment programs in the U.S; 12-step programs recognize and celebrate abstinence [34]. When CM approaches have used verbal praise and recognition as reinforcement, the praise has been given alongside tangible incentives [34]. In Vietnam, substance use-related stigma is rooted deeply, PWUD report feeling more accepted within methadone clinics than in their local communities, leading to recognition incentives being more acceptable within methadone clinical settings than in local neighborhoods where patients live [32]. To make these recognition and praise incentives more culturally suitable, interventions to address stigma and improve provider-patient trust and interaction quality must be implemented first [32,35,36].

We found several other incentives that might be useful in methadone clinical settings in low- and middle-income countries like Vietnam, although these incentives did not stand out as the most or least preferable. Most of these incentives are tangible, and cost money at some level. These incentives, such as cash, vouchers for services fees, lottery tickets to win prizes, gifts for family, vouchers for goods, have been used commonly previously [18,19]. But in resource-limited settings, a key question is determining the incentive cost "threshold" at which the CM intervention is cost-effective. We plan to explore the feasibility of these incentives from the perspective of policymakers' perspective in the future. In addition, the preferences in certain subgroups of PWUD must be explored, especially among people in mountainous areas who may have a higher need for interventions (i.e., higher rates of non-adherence and dropout compared to those in metropolitan areas) [14,37]. Conducting such studies could also help determine whether the benefits of CM varied at the site level, as previous research has suggested higher effects of CM in locations with lower socioeconomic status [22].

Our study sample included one-third participants from suburban methadone clinics which may enhance the generalizability of these findings for patients receiving methadone in Vietnam. Age, employment status, educational attainment and other sociodemographic characteristics are broadly consistent with recent studies among patients in methadone treatment in Vietnam, increasing the compatibility of our study sample to others [14,27,38,39]. However, our study sample may have included many easy-to-approach patients as patient selection was initially based on the clinical staff's assessment and data collection was conducted over a short period (one day per study site). As such, selection bias may have occurred if the preferences for CM incentives differed between our study sample and the intended target population. Although we

expected to recruit participants with poor adherence and retention in methadone treatment (our target population, see Participants subsection in Methods), Table 1 shows only a small proportion of participants who reported missing methadone doses during the past 30 days or a history of dropout from methadone. This suggests that our instruction to clinical staff to prioritize selecting this target population may not have been fully effective. Future research may benefit from using more explicit and objective criteria regarding adherence and retention to better identify this target population.

## Conclusions

Financial support or discounts for monthly methadone treatment fees emerged as the most desirable incentives in a contingency management package for people receiving MMT to reinforce adherence and retention in methadone treatment. Take-home methadone and receiving priority coupons for early checkups or counselling were also patients' preferences as the second and third ranks. Patients least preferred recognition or verbal praise at their local, neighborhood or living areas, as well as at their methadone clinics. We recommend future CM interventions may consider using the most highly preferred incentives as rewards to offer in order to reinforce treatment adherence and retention in methadone treatment. These findings suggest possible low-cost contingency management strategies that could inform decision-making in MMT programs.

## Supporting information

**S1 Table. Reasons for non-participation in the study.**
(DOCX)

**S1 File. Raw data of the study.**
(DTA)

## Acknowledgments

We thank research scientists, research assistants and staff as well as staff in the six methadone clinics who provided critical support to conduct this study. We thank all participants who participated in the study.

## Author contributions

**Conceptualization:** Vivian F. Go, William C. Miller.

**Data curation:** Hue Thi Nguyen, Thuyet Thi Phung.

**Formal analysis:** Thuy Thi Dieu Dao.

**Funding acquisition:** Vivian F. Go, William C. Miller.

**Investigation:** Hue Thi Nguyen, Thuyet Thi Phung.

**Methodology:** William C. Miller.

**Project administration:** Hue Thi Nguyen, Thuyet Thi Phung, Van Hai Hoang.

**Software:** Thuy Thi Dieu Dao.

**Supervision:** Van Hai Hoang, Huong Thi Le, Brian W. Pence, Giang Minh Le, Vivian F. Go, William C. Miller.

**Visualization:** Thuy Thi Dieu Dao, William C. Miller.

**Writing – original draft:** Thuy Thi Dieu Dao.

**Writing – review & editing:** Thuy Thi Dieu Dao, Hue Thi Nguyen, Trang Thu Nguyen, Thuyet Thi Phung, Van Hai Hoang, Huong Thi Le, Brian W. Pence, Giang Minh Le, Vivian F. Go, William C. Miller.

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
