## [Decision Letter · Decision Letter 0]

5 Nov 2025

Dear Dr. Dao,

Thank you for submitting your manuscript to PLOS ONE. After careful consideration, we feel that it has merit but does not fully meet PLOS ONE’s publication criteria as it currently stands. Therefore, we invite you to submit a revised version of the manuscript that addresses the points raised during the review process.

We look forward to receiving your revised manuscript.

Kind regards,

Kimberly Page, PhD, MPH

Academic Editor

PLOS ONE

Journal Requirements:

3. Thank you for stating the following in your manuscript:

[Research reported in this publication was supported by the Fogarty International Center, Eunice Kennedy Shriver National Institute of Child Health & Human Development, the National Institute on Drug Abuse, and the National Institute of Mental Health of the National Institutes of Health under Award Number D43 TW011548. The content is solely the responsibility of the authors and does not necessarily represent the official views of the National Institutes of Health.]

[The author(s) received no specific funding for this work.]

4. In the online submission form, you indicated that [All relevant data are available upon request and acceptance by the principal investigators of the study.].

Reviewers' comments:

Reviewer's Responses to Questions

**Comments to the Author**

1. Is the manuscript technically sound, and do the data support the conclusions?

Reviewer #1: Yes

Reviewer #2: Yes

2. Has the statistical analysis been performed appropriately and rigorously?

Reviewer #1: Yes

Reviewer #2: Yes

3. Have the authors made all data underlying the findings in their manuscript fully available?

Reviewer #1: No

Reviewer #2: Yes

4. Is the manuscript presented in an intelligible fashion and written in standard English?

Reviewer #1: Yes

Reviewer #2: Yes

Reviewer #1: This is a well-written report on an innovative study that has potential to be helpful to clinicians, policy-makers, and researchers interested in implementing contingency management (CM) in low-resourced methadone clinics for opioid use disorder. I have been active in CM research for many years and was unfamiliar with the interesting best-worst analysis method used in this study. I enjoyed learning about it and suspect others may as well. I commend the authors on this innovative study and well-crafted report. I have no recommendations for changes.

Reviewer #2: The study aimed to characterize and measure patient preferences for low-cost incentives within a Contingency Management (CM) framework designed to enhance adherence and retention in Methadone Maintenance Treatment (MMT). It specifically addresses the challenge of implementing CM in settings with limited resources, like a low to medium income country as Vietnam. This study provides empirically-driven insights into patient preferences for low cost incentives. However, there are potential selection bias that need to be discussed. While the authors mentioned they prioritized approaching participants with poor adherence or a history of discontinuation, they did not show if this subgroup behave significantly different from a more general population. Also, as a cross-sectional study, it measured stated preference, which may not be directly translated into actual treatment efficacy. It is not clear, if the preferred incentives would improve adherence and retention in the long run, which further evidence should be presented by the authors or reference materials.

**Do you want your identity to be public for this peer review?** For information about this choice, including consent withdrawal, please see our Privacy Policy

Reviewer #1: No

Reviewer #2: No

---

## [Author Response · Author response to Decision Letter 1]

12 Nov 2025

Thank you for circulating reviewers' comments. Please kindly see below our response.

Reviewer 1:

1. This is a well-written report on an innovative study that has potential to be helpful to clinicians, policy-makers, and researchers interested in implementing contingency management (CM) in low-resourced methadone clinics for opioid use disorder. I have been active in CM research for many years and was unfamiliar with the interesting best-worst analysis method used in this study. I enjoyed learning about it and suspect others may as well. I commend the authors on this innovative study and well-crafted report. I have no recommendations for changes.

Response: We thank the reviewer for the review and support.

Reviewer 2:

1. The study aimed to characterize and measure patient preferences for low-cost incentives within a Contingency Management (CM) framework designed to enhance adherence and retention in Methadone Maintenance Treatment (MMT). It specifically addresses the challenge of implementing CM in settings with limited resources, like a low to medium income country as Vietnam. This study provides empirically-driven insights into patient preferences for low cost incentives. However, there are potential selection bias that need to be discussed. While the authors mentioned they prioritized approaching participants with poor adherence or a history of discontinuation, they did not show if this subgroup behave significantly different from a more general population.

Response: We thank the reviewer for the insightful comment. As we conducted this study to explore patients’ preferred incentives to reinforce retention and adherence, we prioritized choosing participants with this target behavior. We clarified this point in “Participants” subsection (page 7) and as follows:

“Although poor adherence or previous discontinuation of methadone was not an explicit inclusion criterion, the target population we aimed to reach out consisted of patients currently receiving methadone who demonstrated poor adherence and/or were at high risk of treatment discontinuation. These aspects were target behaviors for the CM interventions. We worked closely with clinical staff to identify and approach this population as they presented in the study sites." In this target population, we anticipate a minimal impact of selection bias with less than 10% of non-participation (S1 Table).

However, the selection of patients with poor adherence might limit our generalizability to all patients in methadone, thus, we have a few sentences in the last paragraph of Discussion to talk about potential generalizability.

2. Also, as a cross-sectional study, it measured stated preference, which may not be directly translated into actual treatment efficacy. It is not clear, if the preferred incentives would improve adherence and retention in the long run, which further evidence should be presented by the authors or reference materials.

Response: We thank the reviewer for the important point. We added some sentences in Introduction (please refer to Page 5) for more elaboration on this point. Also, please see below our brief explanation.

As we laid out in Introduction, evidence indicated that the effectiveness of CM on treatment adherence and retention have been seen in both high and low/middle income countries [18,21,22]. But the implementation of CM in low/middle income countries faces particular challenges (which can hinder the CM’s efficacy) [21]. That said, we need to explore ways to strengthen the implementation of CM to ensure its effectiveness in low-income, real-world settings. To improve the implementation, some upstream implementation outcomes, such as acceptability and feasibility, were particularly appropriate to focus on at the beginning. Thus, understanding patients’ preferences has the potential to enhance acceptability and implementation of CM intervention, which ultimately ensures the effectiveness of CM (this effectiveness has been shown in previous studies, some examples as reference 18, 21 and 22).

But we totally agree with the reviewer that in the long run, further research with a focus on the relationship between these types of incentives and longer-term retention and adherence in methadone treatment are needed.

---

## [Editor Report · Decision Letter 1]

11 Dec 2025

Dear Dr. Thuy Dao,

Thank you for submitting your revised manuscript to PLOS ONE. After careful consideration, we feel that it has merit but does not fully meet PLOS ONE’s publication criteria as it currently stands. Therefore, we invite you to submit a revised version of the manuscript that addresses the points raised during the review process.

As Academic Editor, there are a few areas where your responses were not quite addressing what the reviewers requested. I hope I can clarify this so you can resubmit this very interesting paper.  First, Reviewer #2 requested that you address potential selection bias that could arise from your participant recruitment and selection process for the study. Your response, while providing some insight into the process did not fully address this very important potential methodological issue.  The reviewer specifically requested that you make some comparisons between your study sample and broader population in MMT program.  In what ways were they different or not.  This information should be presented briefly in results. This can inform your consideration of how much potential there is for selection bias or not in your sample and how that may or may not have influenced your results and conclusions. Please consider the following:  1) fully describe the inclusion an exclusion criteria.  If staff were directed to 'select' or 'approach' people based on un-documented inclusion or exclusion criteria, this needs to be further elaborated on. This kind of subjective selection can have significant impacts on your results, interpretations, and conclusions.  Please address this more fully with information about who did the approach for the study and how they decided if someone was eligible. Please state in the Results how many people were approached to participate, and also, please state how many and the proportion of those approached who agreed to participate.   2) Please state the dates  during which the study was conducted.3) Please add information about who consented the participants, and was it written? Was participation anonymous, or were names collected? 4) ABSTRACT:  your conclusion paragraph seems to merely reiterate the results.  Can you elaborate more on how these results might inform implementation ?

Please submit your revised manuscript by Jan 25 2026 11:59PM. If you will need more time than this to complete your revisions, please reply to this message or contact the journal office at plosone@plos.org . A rebuttal letter that responds to each point raised by the academic editor and reviewer(s). You should upload this letter as a separate file labeled 'Response to Reviewers'.A marked-up copy of your manuscript that highlights changes made to the original version. You should upload this as a separate file labeled 'Revised Manuscript with Track Changes'.An unmarked version of your revised paper without tracked changes. You should upload this as a separate file labeled 'Manuscript'.

We look forward to receiving your revised manuscript.

Kind regards,

Kimberly Page, PhD, MPH

Academic Editor

PLOS One
---

## [Author Response · Author response to Decision Letter 2]

23 Dec 2025

Please consider the following:

1. fully describe the inclusion and exclusion criteria. If staff were directed to 'select' or 'approach' people based on un-documented inclusion or exclusion criteria, this needs to be further elaborated on. This kind of subjective selection can have significant impacts on your results, interpretations, and conclusions. Please address this more fully with information about who did the approach for the study and how they decided if someone was eligible. Please state in the Results how many people were approached to participate, and also, please state how many and the proportion of those approached who agreed to participate.

Response: We thank the editor for the clarification and comment. We elaborated on the process of selection in participants subsection (Page 8-9)*, then discussed further on the risk of selection bias in Discussion (Page 23-24).

We clarified in Results (Page 14) regarding the number of patients who were approached. As they were first verbally approached by clinical staff, then patients who were interested in the study were referred to research assistants to do formal eligibility screening, introduction and informed consent. We did not document the total number of patients verbally approached by clinical staff; but we documented the number of patients after they met with research assistants, who declined to participate.

2. Please state the dates during which the study was conducted.

Response: We thank the editor for requesting more information. We added time period for data collection in the study design section (Page 6) which was from Nov 14 to 26, 2023 with one day spent collecting data at one site (i.e., 6 days for 6 sites in total).

3. Please add information about who consented the participants, and was it written? Was participation anonymous, or were names collected?

Response: We thank the editor for this comment. We added a paragraph on the procedure when informed consent form was obtained (Page 11). We added the confidentiality aspect in Page 9 where we mentioned IRB approval.

4. ABSTRACT: your conclusion paragraph seems to merely reiterate the results. Can you elaborate more on how these results might inform implementation?

Response: We thank the editor for the comment. We revised and clarified the abstract conclusion. We suggest it would be beneficial for the future contingency management (CM) intervention to consider or start with the most preferred incentives as their first-line rewards to offer in the CM interventions.

*all pages are referred to the “Revised manuscript with track changes” version.

---

## [Editor Report · Decision Letter 2]

6 Jan 2026

Patient Preferences for Incentives in Contingency Management Interventions in Methadone Treatment: A best-worst scale Analysis

PONE-D-25-41128R2

Dear Dr. Dao,

We’re pleased to inform you that your manuscript has been judged scientifically suitable for publication and will be formally accepted for publication once it meets all outstanding technical requirements.

Kind regards,

Kimberly Page, PhD, MPH

Academic Editor

PLOS One

Additional Editor Comments (optional):

The responses to critiques have clarified the study procedures. Thank you for your concise responses!
---

## [Editor Report · Acceptance letter]

PONE-D-25-41128R2

PLOS One

Dear Dr. Dao,

I'm pleased to inform you that your manuscript has been deemed suitable for publication in PLOS One. Congratulations! Your manuscript is now being handed over to our production team.

Kind regards,

on behalf of

Dr. Kimberly Page

Academic Editor

PLOS One